# Measuring Violence Behaviours in the Context of Romantic Relationships during Adolescence: New Evidence about the Modified Conflict Tactics Scale

**DOI:** 10.3390/children10020297

**Published:** 2023-02-03

**Authors:** Javier Ortuño-Sierra, Natalia Marugán Garrido, Andrea Gutiérrez García, Ana Ciarreta López, Tomas Camara-Pastor

**Affiliations:** Educational Sciences Department, University of La Rioja, 26004 Logroño, Spain

**Keywords:** M-CTS, adolescents, psychometric properties, intimate partner violence

## Abstract

The main purpose of the present study was to analyze the psychometric properties of the Spanish version of the Modified Conflict Tactics Scale (M-CTS) in adolescents. The M-CTS is a questionnaire that screens for intimate partner violence. Additionally, we studied the association between the M-CTS and attitudes towards violence. The study included a sample of 1248 students in a cross-sectional survey. The M-CTS and the Attitudes Towards Violence (EAV) scale were used. The analysis of the internal structure of the M-CTS revealed that a four-factor structure was the best fitting solution. The M-CTS scores revealed evidence of structural equivalence by gender and age. The McDonald’s Omega indices were adequate for both victims and perpetrators models. Moreover, attitudes towards violence were positively correlated with measures of violence manifestations. Results found in the present study confirm the psychometric adequacy of the M-CTS scores and gather new evidence about its internal structure and measurement equivalence for its use in samples of adolescent and young students. The assessment of intimate partner violence may contribute to detect adolescents at risk for different forms of violence in the future.

## 1. Introduction

The analysis of violent behaviors in the context of interpersonal relationships during adolescence have sharply increased in recent years [1,2,3,4]. These kinds of behaviors are of a crucial relevance as they may predict future dating violence during adulthood [5,6,7]. Recent research confirms that violent behaviors, including physical, psychological, and sexual violence, are prevalent in the context of romantic relationships [8,9]. Different studies indicate that the prevalence of adolescents admitting that they have acted in some type of violence towards their partners varies from 10% to 50% [10]. On the other hand, the percentage of adolescents indicating that they have suffered some kind of violence related to intimate partner relationships varies from 12% to 45%. Thus, intimate partner violence has become a serious issue during adolescence [11,12,13]. This phenomenon is especially worrying considering that some mental healthproblems starting at this moment of life may perpetuate to adulthood [14,15]. In addition, intimate partner violence is, at present, a global issue affecting women and men.

The modified version of the Conflicts Tactics Scale (M-CTS) [14] is one of the most used instruments to screen for arguments related to romantic relationships and is derived from the original Conflicts Tactics Scale (CTS) [16]. The original CTS was composed of 80 items and five different dimensions (Negotiation; Psychological aggression; Physical assault; Sexual coercion; and Injury). The original version revealed adequate evidence of validity and reliability of the scores from the test. The revised version of this instrument, the CTS2, was composed of 39 items, and from the CTS2, another test, the CTS2S was developed as a 20 items test. [17]. The M-CTS consists of 18 bidirectional items that address both perpetrator and victim in a Likert-type response format with five options. Compared with the original CTS, the M-CTS introduced two different items (have you tried to physically restrain your partner and have you hit your partner), the time setting is for a current or a former relationship, and moreover, the Likert scale ranges from 1 to 5. The items intend to measure the extent to which different manifestations of verbal (e.g., you have argued in a specific way), psychological (e.g., you have cried as a consequence of a discussion), or physical aggression (e.g., you have thrown an object at your boyfriend/girlfriend) appear in the context of a romantic relationship. It is worth noting the M-CTS has also been used in other contexts besides intimate partner relationships, such ascaregivers of people with dementia [18]. Although it is widely used, there are few studies regarding its psychometric adequacy in adolescents. For instance, different studies have shown that the M-CTS has adequate evidence of validity in clinical samples and amongst university students [19,20,21,22]. The M-CTS was translated into Spanish by means of back forward translation, revealing good psychometric properties in a large sample of adolescents and young people from 16 to 26 years old [23], indicating that the M-CTS was composed of four different factors: arguments, psychological/verbal aggression, slight physical aggression, and severe physical aggression. Authors of a Spanish study already indicated in their paper that the necessity of new studies analyzing the internal consistency of the instrument as the argumentation factor did not reach adequate levels of internal consistency of the scores. This, in addition to the fact that relevant aspects such as evidence of the measurement invariance (MI) across variables such as gender or age were not studied indicates the necessity of new studies about the psychometric properties of the Spanish version. If MI does not hold, inferences derived from comparison across variables could be unfounded [24]. More recently, the psychometric adequacy of the M-CTS was studied in a sample of Mexican adolescents [25]. The study, with some modification of six items, found that the four-factorial structure was satisfactory. However, the MI was not studied, and in fact, only a few studies have analyzed the MI of the instrument [21,26]. Therefore, the M-CTS is a relevant instrument for the screening of inadequate behaviors in the context of intimate partner relationships. Additionally, there is a lack of studies about the psychometric properties of the instruments in its Spanish version in samples of adolescents.

Considering this previous background, the main objective of the present study was to analyze the psychometric properties of the Spanish version of the M-CTS in adolescents. We thus (a) studied the evidence about its internal structure through confirmatory factor analysis (CFA); (b) obtained evidence of the MI of the instrument by relevant variables such as gender and age; (c) studied the internal consistency of the scores; and (d) analyzed evidence of the validity of the instrument with regard to external variables. We hypothesized that a four-dimensional model would reveal adequate goodness-of-fit indices. We further hypothesized that this dimensional structure would be invariant across gender and age. In addition, we hypothesized that the M-CTS’ scores would reveal adequate levels of internal consistency and that a positive association would be found between the M-CTS dimensions and attitudes towards violence.

## 2. Materials and Methods

### 2.1. Participants 

The study was conducted in Navarra and La Rioja. We used a convenience sampling method. Participants were recruited from different types of secondary schools (e.g., public, funded, and private) and vocational/technical schools. We selected participants from different cities and different socioeconomic levels as well as rural and urban areas were included (see Table 1). Students belonged to ten different schools, including educational and training centers. Initially, a total of 1305 students were included in the study from a total estimated population of 15,500 students. Participants that did not report information about demographic characteristics, that did not respond to all the items (*n* = 37), and that were considered as outliers (*n* = 20) (e.g., a score over 2.5 standard deviations in the different measures) were removed. Finally, a total sample of 1248 students, of which 483 were men (38.7%) and 765 were women (61.3%), were included in the study. Participants’ age ranged from 13 to 21 years old (M = 16.12 years; SD = 2.12). Age distribution was as follows: 13 years old (*n* = 65; 5.2%), 14 years old (*n* = 216; 17.3%), 15 years old (*n* = 336; 26.9%), 16 years old (*n* = 231; 18.5%), 17 years old (*n* = 147; 11.8%), 18 years old (*n* = 129; 10.3%), and 19–21 years old (*n* = 92; 7.3%).

### 2.2. Instruments

#### 2.2.1. The Modified Conflict Tactics Scale (M-CTS)

The M-CTS [26] is a widely used instrument developed to screen for perpetration and victimization of psychological and physical violence in the context of intimate partner relationships. The M-CTS is composed of 18 bidirectional items with a 5-point response format, ranging from 1 (*never*) to 5 (*very often*). The answer frame of the question refers to their last relationship if at the moment of the test the respondent is not currently in a relationship. The validated Spanish version was used in the present study [23]. Alpha values in the Spanish version were acceptable in all of the subscales besides argumentation (0.315 for perpetrators and 0.306 for victims). 

#### 2.2.2. The Attitudes Scale towards Intimate Violence (Escala de Actitudes hacia la Violencia Íntima, EAV)

The EAV [27] addresses attitudes towards violence in the context of intimate and romantic relationships. The EAV encompasses 10 items in a Likert response format. The response options for each question range from 1 = totally disagree to 5 = totally agree. The EAV intends to address whether the individual thinks that the use of different types of violence is justified in a romantic relationship. For instance, the questionnaire asks it is appropriate to use violence in the following cases: “When a member of the couple insults the other” or “When one member of the couple does not agree to have sexual intercourse”. Previous studies have indicated adequate internal consistency of the EAV scores, revealing alpha values higher than 0.90 [27]. 

### 2.3. Procedures

The research took place during regular school hours. The classrooms were prepared for the research. We applied the questionnaires in group of no more than 35 students per classroom. Informed consent of those participants under 18 was signed by the father, the mother, or the legal tutor of the student. A trained researcher informed participants about the confidentiality of the research. In addition, participants could abandon the research at any moment for any reason. Participants were informed that they could leave the research at any moment and that they would not receive any reward for their participation. Additionally, the trained researcher indicated that the research was about different indicators of wellbeing and that their responses would be anonymous. The ethic committee of the University of La Rioja approved the study.

### 2.4. Data Analyses

First, we analyzed the internal consistency of the M-CTS by means of the McDonald’s Omega. Second, with the aim to study the internal structure of the EAV, we conducted several confirmatory factor analyses (CFA). We tested a one-dimensional factor model, the original four-factor model, and a bifactor solution with a general factor, and four different group factors. We chose the WLSMV estimator for dichotomous items. Different goodness-of-fit indices were considered to assure the model fit: Chi-square (χ^2^), Comparative Fit Index (CFI), Tucker–Lewis Index (TLI), Root Mean Square Error of Approximation (RMSEA), and Weighted Root Mean Square Residual (WRMR). Hu and Bentler [28] proposed RMSEA values under 0.06 as adequate, and CFI and TLI about 0.95 or more. Nonetheless, values over 0.90 are usually considered as acceptable. With regards to WRMR, values less than 0.95 indicate good model fit (for dichotomous outcomes) [29]. Third, we analyzed MI across gender and age by means of multigroup CFAs and attending to delta parameterization. In order to do so, first we studied the configural and then the strong model [24]. With regards to study age, we established two different groups: younger adolescents (13–15 years old) and older adolescents (16–21 years old). To test for MI, we first determined the multigroup baseline model and then we established successive equivalence constraints in the model parameters across the groups. Considering the sensitivity of the ∆χ 2 to sample size, Cheung and Rensvold [30] suggested the change in CFI (∆CFI) as a more accurate criterion to determine if nested models are practically equivalent. Finally, to analyze the correlation between the M-CTS scores and other indicators, including the EAV subscales, we analyzed Pearson’s correlations for quantitative measures. A significance level of α = 0.05 was considered. We used SPSS 24.0 [31] and Mplus 7.4 [32] to analyze the data.

## 3. Results

### 3.1. Descriptive Statistics and Study of the Internal Consistency of the M-CTS Scores

First, we calculated descriptive statistics for the subscales and total scores of the M-CTS. With the aim to calculate the reliability of the scores, the McDonald’s Omega was computed (see Table 2). The results indicated adequate values over 0.80 in all of the M-CTS dimensions. In the case of the dimensions for perpetrators, McDonald’s Omega values ranged from 0.83 (Severe Physical Aggression) to 0.90 (Psychological Aggression). For the M-CTS dimensions for victims, values ranged from 0.82 (Argumentation) to 0.88 (Severe Physical Aggression).

### 3.2. Evidence of Internal Structure

We performed different CFAs at the item level (see Table 3). The goodness-of-fit indices for the different factors’ models are shown in Table 2. As can be seen, the one-dimensional model revealed poor values for the CFI and TLI, both for the victims and perpetrators. We then studied the adequacy of the proposed four-factor model. The goodness-of-fit indices for this model were adequate for both victims and perpetrators, with CFI and TLI values around 0.95. In addition, the RMSEA values were under the 0.06 recommended cut off point, and the WRMR values were under 1. Considering the adequacy of the four-factor model and the fact that the bifactor model displayed lower goodness-of-fit indices both for victims and perpetrators, with some values (e.g., CFI = 0.899) under the recommended cut-off values, we decided to retain the four-factor model as the most suitable model.

### 3.3. Study of Structural Equivalence of the M-CTS Scores by Gender and Age

Once the four-factor model was found as the most satisfactory model, we consequently studied the MI of the M-CTS scores by gender and then age. First, we studied if the four-factor model had a good fit attending to the different groups (see Table 4). The study of goodness-of-fit indices revealed a good adequacy of the models. The configural invariance model indicated an adequate fit to the data for both victims and perpetrators. Then, we studied the strong invariance model by means of constraining items’ thresholds and factor loadings to be equal across groups. The ΔCFI was under 0.01, revealing strong MI by gender and age for the M-CTS scores for both victims and perpetrators.

### 3.4. Evidence of Relation with Other Variables

We studied the association of the EAV total score and the M-CTS subscales by means of Pearson’s correlations. As can be seen in Table 5, the Pearson’s correlation revealed a positive and significant correlation between the measures EAV with regards to psychological aggression as a victim and all the physical aggression both as a victim and as a perpetrator. 

## 4. Discussion

Intimate partner violence is nowadays a global issue that is affecting not only individuals, but families and society as a whole [3,4,33]. In addition, this kind of violence is starting earlier during adolescence, a relevant period where these behaviors develop [11,12,13]. Thus, it is not surprising that intimate partner violence has increased in recent years, becoming a world health issue [1]. However, there are still relatively few studies analyzing attitudes related to romantic relationship violence across the world.

Therefore, this work intended to study the prevalence and psychometric properties of the M-CTS in adolescents and youth students. The results revealed that the M-CTS is an instrument with adequate evidence of validity and reliability of the scores. Thus, the instruments can be used in educational settings such as schools or universities, as well as clinical practice. It is crucial to implement early detection strategies and to promote positive attitudes that could potentially prevent these manifestations. 

The results of the present study revealed adequate psychometric properties of the M-CTS. The internal consistency of the scores, estimated by means of McDonald’s Omega, was good for both victims and perpetrators. In addition, the analysis of evidence about the internal structure of the instrument indicated that a four-factor structure best fits the data. Moreover, this structure was equivalent by gender and age, after the study of the MI. To date, no previous studies have analyzed the M-CTS scores in a non-clinical sample of Spanish adolescents, and only a few studies have analyzed the MI of the instrument [21,26]. For instance, the study of Nocentini et al. [21] found MI by gender for the four-factor structure. Therefore, future studies should further study the extent to which results found in the present work are similar in other samples. A previous study [25] analyzing the internal structure of the M-CTS in a sample of Mexican adolescents revealed similar results to those found in our study. The four-factor structure was the best dimensional model and the internal consistency of the scores was adequate. However, evidence about the measurement equivalence of the questionnaire was not studied. Additionally, this study used a modified version of the Spanish version, where six items were adapted for its use in Mexico. The study carried out by Muñoz-Rivas et al [23], in a large sample of Spanish adolescents, revealed that the four-factor structure was the best dimensional model, although some modifications were needed before reaching adequate goodness-of-fit indices. Therefore, new empirical data are still necessary.

The association between intimate partner violence, measured by means of the M-CTS, and attitudes towards violence by means of the EAV was also studied. Despite this being the first study analyzing the association between the EAV and the M-CTS, previous studies have analyzed the correlation between indicators of attitudes and ideas about violence and outcomes related to gender violence [9,34]. In the present study, the results revealed that the M-CTS scores including victims and perpetrators were statistically significant and positively correlated with the EAV scores. These results are somehow similar to other studies indicating that attitudes toward violence and explicit intimate partner violence were associated [35]. The results seem to confirm the idea that the justification of violence is related to perpetrating violent behaviors in the context of romantic relationships. Adolescence is a key developmental period where attitudes, values, and identity are established. If we aim to prevent intimate partner violence, it seems reasonable that strategies should focus on this stage of development. 

The results of the present study should be interpreted in light of the following limitations. First of all, we used solely self-reporting measures. This means that there are some inherent problems in the interpretation of the scores. An example being the possibility of misunderstanding some items or questions or the lack of introspection of some participants. Thus, future studies using external informants, interviews, or even bio-behavioral and/or biological markers could add valuable information. Second, we did not gather information about other possible psychiatric conditions that may affect the interpretation of the results. Finally, the cross-sectional nature of the study limits the possibility of establishing cause–effect associations.

Notwithstanding these limitations, the results of our study confirm the adequate psychometric properties of the M-CTS, an instrument that assesses intimate partner violence, for its use in adolescent populations. Moreover, these violent manifestations are related to attitudes towards violence during adolescence. The results have clear implications for the use of a relevant instrument such as the M-CTS in school settings, in order to screen for a problem that is becoming more and more relevant in recent years. Moreover, the present work is relevant to better comprehend the structure of intimate partner violence, which can contribute to the implementation of prevention strategies during adolescence. Nonetheless, we still need more studies and information that enhance our comprehension of violence manifestations during adolescence. In addition, the study of MI of the M-CTS attending to variables such as race or culture may improve the information about the structural equivalence of the M-CTS for its use in cross-cultural comparisons. Moreover, the role that intimate partner violence plays during adolescence in the manifestation of other more severe forms of violence during adult relationships could be worthy of analysis by means of longitudinal studies that allow for establishing cause–effect relationships.

## Figures and Tables

**Table 1 children-10-00297-t001:** Demographic characteristics of the sample.

Descriptive Variables	*N*	Percentage	Mean
Age			16.12
13	65	5.2	
14	216	17.3	
15	336	26.9	
16	231	18.5	
17	147	11.8	
18	129	10.3	
19–21	92	7.3	
Gender			
Man	483	38.7	
Woman	763	61.3	
Education level			
Compulsory level	843	67.55	
Baccalaureate	182	14.58	
Vocational/technical school centers	223	17.87	
Region			
La Rioja	850	68.11	
Navarra	398	31.89	
Type of School			
Public	852	68.27	
Funded	334	26.77	
Private	62	4.96	

**Table 2 children-10-00297-t002:** Descriptive statistics for the Modified Conflict Tactics Scale (MCTS) and the Attitudes Scale towards Intimate Violence (Escala de Actitudes hacia la Violencia Íntima, EAV) for the total sample and across gender.

	Total Sample		Men		Women		McDonalds’s Omega
	Mean	SD	Mean	SD	Mean	SD	
**M-CTS**							
Argumentation Victim	8.25	2.71	8.16	2.79	8.30	2.66	0.82
Argumentation Perpetrator	7.87	2.55	7.83	2.69	7.89	2.47	0.85
Psychological Aggression Victim	11.81	4.17	10.29	3.46	12.65	4.29	0.87
Psychological Aggression Perpetrator	11.02	3.78	10.48	3.73	11.32	3.77	0.90
Medium Physical Aggression Victim	8.53	3.25	8.26	3.06	8.68	3.35	0.83
Medium Physical Aggression Perpetrator	8.31	2.98	8.49	3.39	8.21	2.72	0.85
Severe Physical Aggression Victim	3.14	0.99	3.25	1.44	3.09	0.66	0.88
Severe Physical Aggression Perpetrator	3.11	0.77	3.11	0.83	3.11	0.74	0.83
**EAV_TOTAL**	13.99	6.94	14.22	6.95	13.85	6.94	0.87

**Table 3 children-10-00297-t003:** Goodness-of-fit indices for the hypothetical models tested and measurement invariance across gender and age for victims.

Model	χ^2^	*df*	CFI	TLI	RMSEA (90% IC)	WRMR	ΔCFI
One-factor	612.869	67	0.804	0.812	0.093 (0.090–0.102)	2.367	
Four factor model	243.264	63	0.946	0.941	0.043 (0.039–0.047)	0.656	
Bifactor model	840.458	46	0.901	0.895	0.091(0.086–0.097)	2.834	
*Measurement Invariance* *(Four factor model)*							
Men (*n* = 483)	239.082	35	0.950	0.940	0.043 (0.040–0.049)	0.582	
Women (*n* = 765)	287.459	35	0.944	0.939	0.043 (0.041–0.050)	0.598	
Configural invariance	345.515	71	0.953	0.945	0.038 (0.035–0.043)	0.235	
Strong invariance	310.345	107	0.949	0.940	0.042 (0.038–0.045)	0.343	−0.01
13–16 years old (*n* = 953)	380.145	35	0.955	0.948	0.042 (0.038–0.048)	0.310	
17–21 years old (*n* = 295)	398.065	35	0.950	0.946	0.042 (0.036–0.046)	0.305	
Configural invariance	240.689	71	0.956	0.949	0.041 (0.037–0.044)	0.352	
Strong invariance	310.436	107	0.951	0.940	0.040 (0.035–0.043)	0.398	−0.01

*Note*. χ^2^ = Chi square; *df* = degrees of freedom; CFI = Comparative Fit Index; TLI = Tucker–Lewis Index; RMSEA = Root Mean Square Error of Approximation; IC = Interval Confidence; WRMR = Weighted Root Mean Square Residual; ΔCFI = Change in Comparative Fix Index.

**Table 4 children-10-00297-t004:** Goodness-of-fit indices for the hypothetical models tested and measurement invariance across gender and age for perpetrators.

Model	χ^2^	*df*	CFI	TLI	RMSEA (90% IC)	WRMR	ΔCFI
One-factor	512.869	67	0.794	0.801	0.096 (0.090–0.104)	2.497	
Four- factor	341.268	63	0.938	0.931	0.045 (0.039–0.048)	0.956	
Bifactor	865.258	46	0.899	0.898	0.090(0.086–0.096)	2.534	
*Measurement Invariance* *(Four-factor model)*							
Men (*n* = 483)	259.080	35	0.948	0.945	0.043 (0.040–0.049)	0.682	
Women (*n* = 765)	297.479	35	0.943	0.940	0.043 (0.041–0.050)	0.558	
Configural invariance	325.510	71	0.942	0.939	0.044 (0.040–0.050)	0.735	
Strong invariance	330.347	107	0.940	0.936	0.044 (0.038–0.049)	0.743	−0.01
13–16 years old (*n* = 953)	380.126	35	0.936	0.938	0.046 (0.040–0.051)	0.800	
17–21 years old (*n* = 295)	338.065	35	0.950	0.951	0.042 (0.036–0.046)	0.705	
Configural invariance	340.687	71	0.947	0.944	0.041 (0.037–0.044)	0.802	
Strong invariance	360.435	107	0.941	0.940	0.042 (0.039–0.045)	0.898	−0.01

*Note*. χ^2^ = Chi square; *df* = degrees of freedom; CFI = Comparative Fit Index; TLI = Tucker–Lewis Index; RMSEA = Root Mean Square Error of Approximation; IC = Interval Confidence; WRMR = Weighted Root Mean Square Residual; ΔCFI = Change in Comparative Fix Index.

**Table 5 children-10-00297-t005:** Pearson’s correlations between the Modified Conflict Tactics Scale (MCTS) and the Attitudes Scale Towards Violence (EAV).

	1	2	3	4	5	6	7	8	9
EAV_TOTAL (1)	-								
M-CTS Arg A (2)	0.05	-							
M-CTS Arg B (3)	0.01	0.73 *	-						
M-CTS Psy Aggre A (4)	0.09 *	0.13 *	0.19 *	-					
M-CTS Psy Aggre B (5)	0.05	0.19 *	0.23 *	0.76 *	-				
M-CTS Med Phy Aggre A (6)	0.25 *	−0.03	0.01	0.35 *	0.33 *	-			
M-CTS Med Phy Aggre B (7)	0.17 *	0.01	0.03	0.26 *	0.36 *	0.75 *	-		
M-CTS Sev Phy Aggre A (8)	0.35 *	−0.08	−0.05	0.10 *	0.13 *	0.57 *	0.49 *	-	
M-CTS Sev Phy Aggre B (9)	0.26 *	−0.08	−0.08	0.09	0.10	0.36 *	0.43 *	0.71 *	-

**Note**. M-CTS Arg A Argumentation as Victim; M-CTS Arg B = Argumentation as Perpetrator; M-CTS Psy Aggre A = Psychological Aggression as Victim; M-CTS Psy Aggre A = Psychological Aggression as Perpetrator; M-CTS Med Phy Aggre A = Medium Physical Aggression as Victim; M-CTS Med Phy Aggre B = Medium Physical Aggression as Perpetrator; MCTS Sev Phy Aggre A: Severe Physical Aggression as Victim; MCTS Sev Phy Aggre B: Severe Physical Aggression as Perpetrator. * *p* < 0.05.

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
