# Peer review of "Measuring Violence Behaviours in the Context of Romantic Relationships during Adolescence: New Evidence about the Modified Conflict Tactics Scale"

_children, 2023, doi:10.3390/children10020297_

Round 1

Reviewer 1 Report

Dear authors, the paper is original and present some new evidence, however some points should be improved. Please answer all questions in the attached file.

#Introduction

Line 29: What “recent research” are the authors referring? No citation is available in this part of the sentence.

Line 32: Please remove the space after the word “towards”.

Line 61: Please check the double use of the expression “of the”.

#Material and Methods

Line 73: Please check the uppercase in the word “Different”.

Line 84: Please check the double dot in the end of sentence.

Line 104: In the Procedures the authors briefly explain how the study was conducted. However poor information is available about the sampling process. What is the population size? It is obvious that 1248 students are a large number, however in cross-sectional studies it is important to calculate the minimum sample size, based on the probabilistic methods. Please, could the authors provide more information about it? Again, how was the sampling process? How were the schools and adolescents chosen? The recruitment of participants needs to be explained.

Line 112: The data analysis is well explained; however, no information is available about the significance level α. Is it 0.05 in all analysis? This should be reported.

#Results

General Comment: Please, indicate the reference of the table 1 in the body of the text.

Table 4: Why two significance level were used to present the results? This should be pointed in the method.

#Discussion

Line 192: Please check the double comma.

Line 200: “Considering the prevalence”. Which prevalence are the authors referring?

General Comment: The Attitudes Toward Violence (EAV) is not discussed. Why? In truth, the use of Pearson Correlation is a poor method to measure relation between variables and the authors should discuss about this. It seems that the EAV was secondary in the present study.

Author Response

We would like to thank the reviewers for all the insightful comments and appreciations about the study. We have considered all of them and introduced different modifications to the original version of the manuscript. This, in our opinion has substantially improved the quality of the work.

Reviewer 1

#Introduction

Line 29: What “recent research” are the authors referring? No citation is available in this part of the sentence.

Thank you for this comment. New citations have been added to the sentence

Line 32: Please remove the space after the word “towards”.

This has been changed

Line 61: Please check the double use of the expression “of the”.

We appreciate the suggestions. We have modified this in the new version of the manuscript

#Material and Methods

Line 73: Please check the uppercase in the word “Different”.

This has been changed

Line 84: Please check the double dot in the end of sentence.

This has been changed

Line 104: In the Procedures the authors briefly explain how the study was conducted. However poor information is available about the sampling process. What is the population size? It is obvious that 1248 students are a large number, however in cross-sectional studies it is important to calculate the minimum sample size, based on the probabilistic methods. Please, could the authors provide more information about it? Again, how was the sampling process? How were the schools and adolescents chosen? The recruitment of participants needs to be explained.

Thank you for the reviewer comment. Attending to the suggestion, we have included the following in the new version of the manuscript:

“We used a convenience sampling method. Participants were recruited from different types of secondary schools (e.g., public, funded, and private) and vocational/technical schools. We selected participants from different cities and different socioeconomic levels as well as rural and urban areas were included. Students belonged to ten different schools, including educational and training centers. Initially, a total of 1305 students were included in the study from a total estimated population of 15500 students”

Line 112: The data analysis is well explained; however, no information is available about the significance level α. Is it 0.05 in all analysis? This should be reported.

We have included the significance level in the new version of the manuscript:
“Significance level of α = 0.05 was considered.”

#Results

General Comment: Please, indicate the reference of the table 1 in the body of the text.

This has been added

Table 4: Why two significance level were used to present the results? This should be pointed in the method.

We have modified this consequently to the modifications conducted in the data analysis section

#Discussion

Line 192: Please check the double comma.

Thanks for the appreciation

Line 200: “Considering the prevalence”. Which prevalence are the authors referring?

We have modified and delete this in the new version of the manuscript.

General Comment: The Attitudes Toward Violence (EAV) is not discussed. Why? In truth, the use of Pearson Correlation is a poor method to measure relation between variables and the authors should discuss about this. It seems that the EAV was secondary in the present study.

We agree with the reviewer comment in the idea that the EAV could be further elaborated in the discussion. It is also true that Pearson’s correlation is probably not the most sophisticated method to analyze the correlation between two variables, although is very used in studies of validation, as it is the case of the present study. The following has been added in the new version of the manuscript, attending to the reviewer comment:

“The association between intimate partner violence, measured by means of the M-CTS, and attitudes towards violence by means of the EAV was also studied. Despite this is the first study analyzing the association between the EAV and the M-CTS, previous studies have analyzed the correlation between indicators of attitudes and ideas about violence and outcomes related to gender violence [2,14]. In the present study, the results revealed that the M-CTS scores including victims and perpetrators were statistically significant and positively correlated to the EAV scores. These results are somehow similar to other studies indicating that attitudes toward violence and explicit intimate partner violence were associated [15]. The results seem to confirm the idea that the justification of violence is related to engage in violence behaviors in the context of romantic relationships. Adolescence is a key developmental period where attitudes, values, and identity are established. If we aim to prevent intimate partner violence, it seems reasonable that strategies should focus on this stage of development.”

Reviewer 2 Report

In the manuscript "Measuring violence behaviors in the context of romantic relationships during adolescence: New evidence about the Modified Conflict Tactics Scale" the authors propose to analyze the psychometric properties of the  Spanish version of the (M-CTS) in adolescents. In my opinion, a more accurate description of the tool is missing in the manuscript. It would be interesting to understand more, through some examples, the items that compose it, especially because in the discussions they underline how this tool is applicable in educational settings like school or university, as well as clinical practice. It might also be interesting to briefly introduce the reader to how it has been used in previous studies particularly compared to other constructs considered. In the section of the paper where the procedures have been described, the authors could better specify, because at the moment it seems to be missing, whether the consents have been signed by the parents. In addition, what indications were given to adolescents before the administration of the questionnaire? They are not specified.

Author Response

Reviewer 2

In the manuscript "Measuring violence behaviors in the context of romantic relationships during adolescence: New evidence about the Modified Conflict Tactics Scale" the authors propose to analyze the psychometric properties of the  Spanish version of the (M-CTS) in adolescentes

In my opinion, a more accurate description of the tool is missing in the manuscript. It would be interesting to understand more, through some examples, the items that compose it, especially because in the discussions they underline how this tool is applicable in educational settings like school or university, as well as clinical practice.

We appreciate the reviewer comment about this relevant aspecto f the manuscript. We have introduced more lines about the instrument and examples of some items in the new version of the manuscript:

“The modified version of the Conflicts Tactics Scale (M-CTS) [14], is one of the most used instruments to screen for arguments related to romantic relationships and is the derived from the original Conflicts Tactics Scale (CTS) [3]. The original CTS was composed of 80 items and five different dimensions (Negotiation; Psychological aggression; Physical assault; Sexual coercion; and Injury). The original version revealed adequate evidences of validity and reliability of the scores. The revised version of this instrument, the CTS2 was composed of 39 items, and from that a short form with 20 items, the CTS2S was developed [4]. The M-CTS consist of 18 bidirectional items that address both perpetrator and victim in a Likert-type response format with five options. The items intend to measure the extent to which different manifestations of verbal (e.g. you have argued in a pacific way), psychological (e.g. you have cried as a consequence of a discussion) or physical aggression (e.g. you have thrown an object to your boyfriend/girlfriend) appear in the context of a romantic relationship.”

It might also be interesting to briefly introduce the reader to how it has been used in previous studies particularly compared to other constructs considered.

We appreciate the reviewer comment. We have introduced some Other information about the M-CTS in Other contexto besides intimate partner relationshinps:

“Worth noting the M-CTS has been also used in other context besides intimate partner relationships, like for instance caregivers of people with dementia [5]. In addition, evidences of its psychometric properties have been found in clinical samples, as well as in university students  [6,7]”

In the section of the paper where the procedures have been described, the authors could better specify, because at the moment it seems to be missing, whether the consents have been signed by the parents.

We have clarified this point in the new version of the manuscript

In addition, what indications were given to adolescents before the administration of the questionnaire? They are not specified

Attending to the reviewer suggestion, we have added the following information in the new version of the manuscript:

Participants were informed that they could leave the research at any moment and that they would not receive any reward for their participation. Also, the trained researcher indicated that the research was about different indicators of wellbeing and that their responses would be anonymous”

Reviewer 3 Report

The authors examined the psychometric properties of the Spanish version of the Modified Conflict Tactics Scale among adolescents.

1) 1. Introduction: Please provide the citation for "Recent research confirms that violent behaviors, including physical, psychological, and sexual violence, are prevalent in the context of romantic relationships."

2) 1. Introduction: Before introducing the modified version of the Conflicts Tactics Scale (M-CTS), kindly discuss its original version.

3) 1. Introduction: "This study showed that the M-CTS was composed of 4 different factors: arguments, psychological/verbal aggression, slight physical aggression, and severe physical aggression." Are the authors referring to their own study or another study?

4) 1. Introduction: The readers would expect: (a) How M-CTS different from its original version? (items were modified? fewer items?); (b) why measurement invariance across variables such as gender and age is important to be studied?; and (c) discuss more on the modification of 6 items in a sample of Mexican adolescents (are the authors modifying the same 6 items in their study?)

5) 2.1. Participants: (a) What is the sampling method? (b) Is parental consent obtained? (c) A table that shows the demographic characteristics of participants is needed (e.g., types of secondary schools, cities, socioeconomic levels, rural/urban areas, schools, etc.).

6) 2.2. Instruments: Please (a) give an example item for each subscale; (b) discuss the validated Spanish version (e.g., translation methods, validity and reliability, and why this study is needed considering there is already a validated Spanish version); and (c) reliability coefficient of EAV.

7) 4. Discussion: "Therefore, this work intended to study the prevalence ...". Where are the results regarding prevalence?

8) The reference list is incomplete.

Author Response

Reviewer 3

The authors examined the psychometric properties of the Spanish version of the Modified Conflict Tactics Scale among adolescents.

1) 1. Introduction: Please provide the citation for "Recent research confirms that violent behaviors, including physical, psychological, and sexual violence, are prevalent in the context of romantic relationships."

Thank you for this appreciation. We have modified introduced a citation as suggested by the reviewer

2) 1. Introduction: Before introducing the modified version of the Conflicts Tactics Scale (M-CTS), kindly discuss its original version.

We think that this is a relevant comment and we have consequently introduced some new lines discussing the original version:

“The modified version of the Conflicts Tactics Scale (M-CTS) [14], is one of the most used instruments to screen for arguments related to romantic relationships and is the derived from the original Conflicts Tactics Scale (CTS) [3]. The original CTS was composed of 80 items and five different dimensions (Negotiation; Psychological aggression; Physical assault; Sexual coercion; and Injury). The original version revealed adequate evidences of validity and reliability of the scores. The revised version of this instrument, the CTS2 was composed of 39 items, and from that a short form with 20 items, the CTS2S was developed [4].”

3) 1. Introduction: "This study showed that the M-CTS was composed of 4 different factors: arguments, psychological/verbal aggression, slight physical aggression, and severe physical aggression." Are the authors referring to their own study or another study?

We were referring to the previous mentioned study. Nonetheless, we have modified this to make it less confusing

4) 1. Introduction: The readers would expect: (a) How M-CTS different from its original version? (items were modified? fewer items?);

Compared to the original CTS, the M-CTS introduced two different items (have you tried to physically restrain your partner and have you hit your partner), the time setting is for a current or a former relationship, and moreover, the Likert scale range from 1 to 5. The items intend to measure the extent to which different manifestations of verbal (e.g. you have argued in a pacific way), psychological (e.g. you have cried as a consequence of a discussion) or physical aggression (e.g. you have thrown an object to your boyfriend/girlfriend) appear in the context of a romantic relationship. Worth noting the M-CTS has been also used in other context besides intimate partner relationships, like for instance caregivers of people with dementia [5]. In addition, evidences of its psychometric properties have been found in clinical samples, as well as in university students [6,7].

“(b) why measurement invariance across variables such as gender and age is important to be studied?;

We think that this appreciation is relevant. We have introduced the following in the new version of the manuscript:
If MI does not hold inferences derived from comparison across variables could be unfounded [8]”

and (c) discuss more on the modification of 6 items in a sample of Mexican adolescents (are the authors modifying the same 6 items in their study?)

We did not modify the 6 items mentioned in our study, as the study was conducted in Spain.

The following has been added in the discussion section:

“Also, this study used a modified version of the Spanish version, where 6 items were adapted for its use in Mexico.”

5) 2.1. Participants: (a) What is the sampling method?

We think that the comment is relevant- We have added the following information in the manuscript:
“We used a convenience sampling method. Participants were recruited from different types of secondary schools (e.g., public, funded, and private) and vocational/technical schools. We selected participants from different cities and different socioeconomic levels as well as rural and urban areas were included. Students belonged to ten different schools, including educational and training centers. Initially, a total of 1305 students were included in the study from a total estimated population of 15500 students”

(b) Is parental consent obtained?

Yes, we have clarified this in the manuscript:

Informed consent of those participants under 18 was signed by the father, the mother or the legal tutor of the student.”

(c) A table that shows the demographic characteristics of participants is needed (e.g., types of secondary schools, cities, socioeconomic levels, rural/urban areas, schools, etc.).

We have introduced a new table with the demographic characteristics of the sample

6) 2.2. Instruments: Please (a) give an example item for each subscale;

Attending to the reviewer suggestion, we have added this in the introduction of the new version of the manuscript when describing the instrument:

“The items intend to measure the extent to which different manifestations of verbal (e.g. you have argued in a pacific way), psychological (e.g. you have cried as a consequence of a discussion) or physical aggression (e.g. you have thrown an object to your boyfriend/girlfriend) appear in the context of a romantic relationship.” (b) discuss the validated Spanish version (e.g., translation methods, validity and reliability, and why this study is needed considering there is already a validated Spanish version);

We think that the reviewer comment is relevant. We have introduced some new information to clarify this point in the introduction section:

Authors of the Spanish study already indicated in the paper the necessity of new studies analyzing the internal consistency of the instrument as the argumentation factor did not reach adequate levels of internal consistency of the scores. This, in addition to the fact that relevant aspects like evidence of the measurement invariance (MI) across variables like gender or age were not studied indicates the necessity of new studies about the psychometric properties of the Spanish version. If MI does not hold inferences derived from comparison across variables could be unfounded [8]”

and (c) reliability coefficient of EAV.

The alpha value of the previous study has been added in the instruments section

7) 4. Discussion: "Therefore, this work intended to study the prevalence ...". Where are the results regarding prevalence?

Thank you for the appreciation, we have deleted this, as it was misleading.

8) The reference list is incomplete

We have checked the reference list for possible mistakes

Round 2

Reviewer 2 Report

No comments for the authors